# 'We have been in lockdown since he was born': a mixed methods exploration of the experiences of families caring for children with intellectual disability during the COVID-19 pandemic in the UK

Jeanne Wolstencroft ![ORCID] ,[1] Laura Hull,[2] Lauren Warner,[1] Tooba Nadeem Akhtar ![ORCID] ,[1] William Mandy,[2] IMAGINE-ID consortium, David Skuse[1]

¹The Great Ormond Street Institute of Child Health, University College London, London, England
²Division of Psychology and Language Sciences, University College London, London, England

**Correspondence to**
Dr Jeanne Wolstencroft;
j.wolstencroft@ucl.ac.uk

## ABSTRACT

**Objectives** This study aimed to explore the experiences of parents caring for children with intellectual and developmental disabilities (IDD) during the UK national lockdown in spring 2020, resulting from the COVID-19 pandemic.

**Design** Participants were identified using opportunity sampling from the IMAGINE-ID national (UK) cohort and completed an online survey followed by a semistructured interview. Interviews were analysed using thematic analysis.

**Setting** Interviews were conducted over the telephone in July 2020 as the first UK lockdown was ending.

**Participants** 23 mothers of children with intellectual and developmental disabilities aged 5–15 years were recruited.

**Results** Themes reported by parents included: managing pre-existing challenges during a time of extreme change, having mixed emotions about the benefits and difficulties that arose during the lockdown and the need for appropriate, individualised support.

**Conclusions** Our findings confirm observations previously found in UK parents of children with IDD and provide new insights on the use of technology during the pandemic for schooling and healthcare, as well as the need for regular check-ins.

## INTRODUCTION

Intellectual and developmental disabilities (IDD) affect approximately 1.8% of children worldwide.[1] They are associated with significant limitations in cognitive and adaptive skills.[2] Genetic variation can be identified in up to 60% of IDD cases.[3] These changes can be large chromosomal abnormalities (such as aneuploidies and translocations), submicroscopic deletions or duplications (such as copy number variants (CNV)) or small changes in DNA sequence (such as single-nucleotide variants (SNV)).

The IMAGINE-ID consortium has recruited children with IDD of identified genetic aetiology from across the UK. The COVID-19 pandemic poses unique challenges to this cohort in two respects. First, children with IDD have high levels of pre-existing health conditions and so may be considered especially vulnerable to the virus. As a result, families of these children are more likely than the general population to have been shielding, following UK government advice. Second, children with IDD are more likely to make use of services (such as specialist education, healthcare and social care) which were suspended or diminished during the pandemic.[4–8]

**STRENGTHS AND LIMITATIONS OF THIS STUDY**

⇒ This study recorded the experiences of parents who were caring for children with intellectual and developmental disabilities during the first UK national lockdown.

⇒ Short questionnaires assessed the emotional well-being of parents and their children quantitatively.

⇒ A total of 23 caregivers were interviewed in July 2020 as the first UK lockdown was ending and were asked to reflect on their experiences across the entire lockdown.

⇒ Only mothers were interviewed and so results may not reflect the experiences of other caregivers of children with intellectual and developmental disabilities.

Calls have been made to prioritise the collection of high-quality data on the mental health effects of the pandemic, particularly within vulnerable groups.[9 10]

In 'normal times', children with IDD are at increased risk of behavioural difficulties compared with typically developing children.[11–16] Families of a child with IDD are more likely to live in deprived socioeconomic circumstances[17–21] and experience increased parenting stress or parental psychological distress[22–28] than typically developing children. Early reports of the impact of the pandemic on children with IDD and their parents suggest that many were feeling overwhelmed, forgotten and had experienced high levels of anxiety and depression.[9 29–35]

International surveys and interviews with families of children with IDD or physical disabilities suggest that COVID-19 national lockdowns have led to increased behavioural problems for children,[36] as well as reduced opportunities to socialise.[4] Parents have also reported concerns about their child's safety and that children may fall behind developmentally because of reduced access to services.[37]

The nature and extent of COVID-19 restrictions have varied across different countries, with the UK being particularly severely impacted by infections. It is therefore important to assess the experiences of families in the UK in addition to the emerging international literature. One previous qualitative study has examined the experiences of eight mothers of children with IDD in the UK.[33] These authors found that mothers felt under pressure to provide their child with educational and therapeutic provisions, which had been removed or reduced because of the pandemic. Mothers reported a lack of support from additional services, leading to increased stress and uncertainty, but also enjoyed the freedom from routine with several noting the positive impact it had for their child's well-being.

We interviewed the caregivers of children with IDD, who have been caring for their children at home during the pandemic. The present study was carried out concurrently with those described above and we seek to extend their useful findings in the IMAGINE-ID cohort to identify consistencies across research as well as any additional insights from the parents in our cohort.

## Aims and objectives

We sought to understand the experiences of families caring for a child with IDD during the COVID-19 pandemic in order to identify key targets for support. We interviewed 23 IMAGINE-ID families mid-July 2020. At this time, the lockdown restrictions were eased in parts of the UK, but the government advised vulnerable individuals to continue to shield until the 1 August 2020. We asked families to reflect on their experiences across the entirety of the first national lockdown. Our objectives were: (1) to identify the areas of difficulty and resilience during the pandemic and deconfinement and (2) to identify key targets for support.

## METHODS

### Study design

The IMAGINE-ID programme of research, funded by the UK Medical Research Council from 2014 to 2024, recruited a cohort of 3402 UK families whose child had IDD due to an identified genetic anomaly (www.imagine-id.org). IMAGINE-ID caregivers with children under the age of 16 years were invited to take part through the study newsletter. Parents were enrolled in the current study through consecutive sampling due to the time restrictions, so that all participants were interviewed prior to end of shielding in the first UK lockdown. A mixed methods approach was used to capture the diversity of challenges encountered and resilience factors drawn on by families. Data were collected by parent report using online questionnaires and a semistructured telephone interview (n=23). Questionnaire data were collected and managed using REDCap electronic data capture tools hosted at University College London.[38] Telephone interviews were recorded and transcribed by the research team.

### Patient and public involvement

The research objectives were developed in response to conversations with collaborators at the UNIQUE rare chromosome disorder charity. A summary of the results will be disseminated to participants and the IMAGINE-ID cohort via the study newsletter, website and social media accounts.

### Measures

*COVID-19 impact:* An adapted version of the National Institute of Health CoRonavIruS Health Impact Survey was used to assess the family's adaptations to the pandemic, including key domains relevant to mental distress and resilience.[39]

*Child characteristics:* Demographic details and genetic data about the child were extracted from the IMAGINE-ID database. The Wessex scales were used to gain an updated measure of adaptive function and degree of IDD.[40] The measure assesses self-help skills, literacy, mobility and incontinence.

*Well-being:* The well-being of the child and the parent were assessed using the Strengths and Difficulties Questionnaire (SDQ) and the Hospital Anxiety and Depression Scale (HADS), respectively. The SDQ is a well-validated behavioural screening questionnaire for children and has been validated with the parents of children with disabilities.[41] The SDQ includes scales that measure emotional symptoms, conduct problems, hyperactivity/impulsivity and inattention difficulties, peer relationship problems and prosocial behaviour. The first four scales are combined to create a total difficulties score. Of the total difficulties, scale scores of 17–19 are indicative of 'high' levels of behavioural difficulty and 20–40 are considered

'very high' levels of behavioural difficulty compared with population norms. An additional impact scale measures the impact of this composite score on daily life. Scores of 3–10 on the impact scale are in the 'very high' range compared with population samples. The HADS was employed as a self-report measure of parental mental health.[42] It has been validated for use with the parents of children with disabilities.[43] It includes a total of 14 items, with 7 depression items (eg, 'I feel as if I am slowed down') and 7 anxiety items (eg,' I get sudden feelings of panic'). For the anxiety and depression scales, scores in the 0–7 range are considered 'normal', scores in the 8–10 range are considered 'borderline' and scores in 11–21 range are considered 'abnormal'.

*Interview:* The semistructured interview schedule was developed to examine the experiences of parenting a child with additional needs during the pandemic. It aimed to identify the unique challenges experienced by families as well as resilience factors. Questions invited caregivers to reflect on their experiences of the lockdown, including how their day-to-day life had changed, how they coped and identifying coping strategies they had implemented. Parents were asked about their experiences across the UK lockdown, including reflecting back on their experiences during the period of stricter restrictions, as well as the easing of restrictions taking place at the time of the interview (see online supplemental material 1 for the exact questions asked). Interviews ranged from 13 min to 1 hour and 25 min, with a mean length of 44 min, and all interviews were conducted by JW and LH.

## Analysis

Questionnaire data were summarised using descriptive statistics. Deidentified transcripts of interviews were analysed using reflective thematic analysis as described by Braun and Clarke.[44–46] Analysis was led by JW and LH. Adopting a critical realist framework, analyses involved identifying both semantic and latent meanings in the dataset following an inductive approach, whereby themes were generated in response to interview data, rather than trying to accommodate data within predefined themes or research questions.[45] The lead authors (JW and LH) are research psychologists, whose theoretical stance is grounded in a social model of disability. JW's research background is focused on understanding the impact of rare genetic disorders on the neurodevelopment of children with rare genetic disorders. LH's expertise is in camouflaging in autism, using qualitative and quantitative methods to explore characteristics related to identity, mental health and support. None of the authors have IDD themselves, but there is personal and professional experience of IDD among authors.

JW and LH proceeded through the stages of data familiarisation, coding, theme development and review. Themes were reviewed following several detailed discussions with the other members of the research team DS and WM. Following these discussions, the final themes and subthemes were confirmed. In a final stage, the full

**Table 1** Participant characteristics

| ID | Gender | Age (years) | SDQ |
|----|--------|-------------|-----|
| P1 | M | 9 | Very high |
| P2 | M | 5 | Very high |
| P3 | F | 12 | Close to average |
| P4 | F | 5 | Very high |
| P5 | F | 7 | High |
| P6 | F | 13 | Slightly raised |
| P7 | F | 6 | Very high |
| P8 | M | 13 | High |
| P9 | M | 11 | Close to average |
| P10 | M | 10 | Very high |
| P11 | F | 9 | High |
| P12 | M | 8 | Very high |
| P13 | M | 13 | Close to average |
| P14 | M | 7 | Very high |
| P15 | F | 13 | Very high |
| P16 | F | 10 | Close to average |
| P17 | M | 9 | Very high |
| P18 | M | 15 | Very high |
| P19 | M | 14 | High |
| P20 | M | 12 | Slightly raised |
| P21 | M | 9 | Close to average |
| P22 | F | 9 | High |
| P23 | M | 10 | Very high |

F, female; M, male; SDQ, Strengths and Difficulties Questionnaire.

transcripts were coded by independent reviewers LW and TNA following a codebook to ensure consistency.

## RESULTS
### Situating the sample

A total of 23 parents took part in interviews. All the respondents were the mothers of IMAGINE-ID children, five were single parents. They reported on 14 boys and 9 girls aged 5–15 years (M=9, SD=2.9, table 1). Overall, 78% of children received extra help at school or attended a special educational needs school. Most of the children were reported to be of white British ethnicity (n=19), other ethnicities included one mixed white and black child, one Irish child and one Asian child. The genetic make-up of the children was varied and included 3 children with sex chromosome aneuploidies, 16 with a CNV and 4 with an SNV (online supplemental table 1). All but three of the families included siblings. The Wessex scales indicated that the group included one non-verbal child. Two children were non-ambulant, 15 were partly mobile and the remainder were fully mobile. Seven children were not fully continent and only three children

were reported to be literate. None had severe hearing or sight impairments.

On the SDQ, 39% of children were rated to have behavioural difficulties in the 'high–very high' severity bands. In 91% of children, the behavioural difficulties were rated to have a 'very high' impact on the family's day-to-day life. Overall, 35% of children had been diagnosed with an Autism Spectrum Disorder (ASD) and five had a diagnosis of Attention Deficit Disorder or Attention Deficit/Hyperactivity Disorder (ADHD).

When rating themselves on the HADS for anxiety symptoms, 17% of parents scored in the borderline range and 26% scored in the abnormal range. For depressive symptoms, 22% scored in the borderline range and 4% in the abnormal range.

*COVID-19 impact:* Two families suspected that their household had been exposed to someone likely to have COVID-19, but only one household reported having a member of the household test positive for COVID-19. Two families reported non-household family member deaths due to COVID-19. At the time of interviews, none had a confirmed case in their immediate family.

*Adapting to the restrictions:* A quarter of families were shielding due to concerns about vulnerability to the virus. Overall, 83% reported that their child had had some difficulty in following the recommendations for keeping away from close contact with people and 87% reported that the restrictions on leaving home had been stressful for their child. Overall, 61% of children's schools had been closed.

*Financial stability:* The pandemic had a financial impact on families, with 39% of respondents reporting that the pandemic had reduced their ability to earn money.

*Emotional impact on child:* 65% of children were reported to be worried about becoming infected and 35% were concerned about their physical health. Overall, the children were reported to be more worried about friends and family becoming infected by COVID-19 than themselves.

The pandemic led to a few unexpected positives, with 74% of families reporting that the pandemic had led to some positive changes in their child's life. These will be described in more detail in the qualitative results.

### Thematic analysis
Three main themes were identified, representing a range of experiences across different families. Overall, parents described the difficulty of adjusting to the pandemic with the additional challenge of their child's specific needs ('managing pre-existing challenges'); they described unexpected benefits and unanticipated challenges ('mixed emotions') and they emphasised that personalised support was essential for their families to get through the lockdown ('support matters'; table 2).

### Theme 1: Managing pre-existing challenges, in a time of increased strain for everyone 'you just feel like our life was so different to other people's'
Parents described the unique daily challenges they faced before the pandemic, emphasising how different their lives were to those of families without children with special needs. Some families were surprised to find

**Table 2** Summary of themes and subthemes

| Theme | Subtheme |
|---|---|
| Managing pre-existing challenges, in a time of increased strain for everyone 'you just feel like our life was so different to other people's' | 'We've been in lockdown since he was born' Social distancing is the norm |
| | Left behind: 'I feel like they just left people who are vulnerable behind for two months.' |
| | Planning for complex needs 'there's a lot more things that I needed to do than the average sort of family'<br>▶ Fewer resources, but behavioural issues are the same<br>▶ Explaining covid |
| Mixed emotions around the challenges and unexpected benefits of lockdown: 'The pandemic was nice but really hard' | Desperate for hugs: 'I don't like lockdown because I want snuggles with my nanny' |
| | Happy at home: 'Everyone's keeping distance from me and that's how I like it' |
| | Strained relationships: 'Being constantly 24/7 together definitely did build up pressure' |
| | Spending time together and slowing down: 'It brought us a lot closer together' |
| Support matters | Transition to telehealth 'I don't think you can replace face to face with a telephone' |
| | Considering equality and equity in remote schooling<br>▶ Equality through access to technology: 'I think the first thing to do is make sure everyone is able to access what they're providing'<br>▶ Equity of access lagging behind as adaptations aren't always appropriate |
| | Checking in: 'just being able to have that support bubble, rather than being locked in your own four walls' |

that some of the challenges brought on by the lockdown were familiar.

### Subtheme 1.1: 'We've been in lockdown since he was born' Social distancing is the norm

Many families described that social distancing had been part of their daily life before the pandemic. Some aspects of social distancing, such as going to public places outside of peak hours, avoiding busy social settings and monitoring crowds, felt normal.

> We've been in lockdown since he was born. When we go on holiday we have to think about how busy it's going to be. We have to think about how many people are going to be there, if we're going to a restaurant; is it too noisy? Are there too many people? Are people too close to us? So we've always got social distancing in our heads anyway and we always have done. I wish other people could see (…) So that people with normal families could read it and think we're all going back to our normal life but these families aren't. It's definitely a different world that we live in. (P23)

For some, the social distancing measures did not have an impact on their social life, as they already had limited opportunities to socialise.

> To be honest for me it's not all that different than usual because I don't really see a lot of people anyway. (P4)

But for many it increased their sense of isolation.

> I think you just feel more isolated than you would normally, and it was always a bit like that on summer holiday. So if I go out and see all of the kids off and they're doing things that we could never really ever do, but it was just kind of that times 100, that feeling. (P12)

### Subtheme 1.2: Left behind: 'I feel like they just left people who are vulnerable behind for two months.'

Many parents described feeling left behind by a range of services and felt that their child's well-being was considered less important because of their additional needs.

> I just felt like he was forgotten and left to it, like his education doesn't matter because he has special needs. (P12)

> Like just the support and stuff, obviously for children with special needs hasn't been there. (P14)

Furthermore, many parents believed their needs had not been prioritised. They felt let down in comparison to the support that had been offered to other families.

> I think there is definitely something to say for the fact that our community hasn't been mentioned and how have they been supported. Because I certainly haven't had much. (P20)

### Subtheme 1.3: Planning for complex needs 'there's a lot more things that I needed to do than the average sort of family'

Despite some aspects of lockdown being familiar, many felt the lockdown restrictions had created a new set of challenges to be navigated. Parents described having to manage an increase in behavioural problems, uncertainties in co-ordinating medical care and new worries about explaining COVID-19.

*Fewer resources, but behavioural issues are the same*

Managing behavioural difficulties and challenging behaviour was not uncommon for families prior to the pandemic. However, parents described the strain of managing these existing behavioural concerns with fewer resources, such as respite care, school, social care and their support networks. Some highlighted an increase in intensity and frequency of behavioural outbursts.

> Meltdowns, swearing, shouting, taking his frustrations out on us. On the furniture, chairs being knocked over, doors being slammed. So rather than being able to talk to us and say 'I don't like what's happening,' it came out in meltdowns and fairly intense. (P18)

In some cases, this contributed to the escalation of pre-existing behavioural outbursts and mental health problems, to the point of serious concern.

> I think the main thing that got me, and affected me the most was [child 1] saying, when he got so frustrated that he just wanted to kill himself. (P14)

Some parents noted the reappearance of challenging behaviours that had previously been well managed.

> Yeah he gets very hyperactive, very loud we get a lot throwing. We were working with a child psychologist at school for about 5 years on a lot of his behaviours and a lot that we nipped in the bud and improved have come back definitely in the initial stage of the lockdown. (P10)

Changes in routine were the source of anxiety and behavioural outbursts for many children.

> His normal life completely changed. So the anxiety was more about 'I'm being told I can't do the things I would normally do.' And that then causes anxiety because he still hasn't got control over what he would normally do. (…) He needed to have some routine or some structure around his day. (P18)

### Explaining Covid, changing rules and following rules: 'How do you explain a pandemic?'

The level of understanding of the pandemic in the children varied substantially. Most had some awareness of the virus and the restriction measures, but some were unaware of the pandemic completely.

> I don't think he really understood at all, other than that he wasn't going to school and he wasn't doing what he normally does, but he really didn't know why.

Even though on some levels his understanding is quite good and we can explain things to him. I mean how do you explain a pandemic? (P10)

I think he understands. I asked him yesterday how far two metres is: 'am I two metres away from you?' and he said 'no'. So he's got that sort of understanding. (P17)

Some families found using visual aids and social stories helpful to explain the effects of the pandemic to their child, but there were limits to how much could be conveyed in a simplified way.

We asked for social stories, so he had a social story about washing his hands, about distancing, the fact that there was going to be a one-way system around the school. So by having that, we kind of prepared him for those things. (P18)

Many children had some difficulty applying the social distancing measures to friends and family. This meant that some parents modified their behaviour and avoided certain situations to keep their children safe.

We've been doing shopping trips without him, but when we have sort of bumped into friends when we've been out and about he doesn't quite understand. We have to make sure to keep hold of him because he wants to go and give auntie a cuddle and he doesn't quite understand that he can't do that. (P10)

Some children were reported to understand the rules and were keen to follow them. Some of these children found it difficult to understand why other people were not following the rules in public places and at times attempted to police strangers about their rule-breaking behaviour.

She's very good at rules. She's very interested in the Boris Johnson announcements and she takes them very literally. So if she sees somebody in the street not obeying social distancing law she won't think twice about telling them that: 'you're not social distancing and you need to be this far away'. (P22)

## Theme 2: Mixed emotions around the challenges and unexpected benefits of lockdown: 'The pandemic was nice but really hard'

When reflecting on the overall experience of the pandemic and particularly the lockdown, most parents had mixed emotions. Although there were many challenging moments, for many families there were also unexpected benefits which parents wanted to extend beyond the pandemic.

### Subtheme 2.1: Desperate for hugs: 'I don't like lockdown because I want snuggles with my nanny'

One common report from parents was that children were strongly impacted by not being able to see friends or extended family. Children missed both physical and emotional contact with others and communicated that to their parents.

He talks about the school and people he misses, so I know he does that when he misses friends or family – he mentions their name or with a question mark at the end quite clearly in his voice, that he's missing them and wants to see them. (P20)

Some children were also frustrated by not being able to take part in activities they normally enjoyed, whether because the activities were cancelled or because the family was shielding.

He's a nature boy he likes to go outside of the house, he loves going to places where there are animals or somewhere with a stream you know or that sort of thing, so he's sort of in his element there, so I think he's definitely missing out. (P2)

### Subtheme 2.2: Happy at home: 'Everyone's keeping distance from me and that's how I like it'

However, quite a few parents reported that their child with IDD was happy being at home during the lockdown. Many children felt that home was a safe and comforting place, where they could follow their own preferred routines and did not have to worry about the anxieties of school or the outside world.

She loved being in, she'd rather be at home any day. She'd shut herself away and she plays on her own a lot, so yeah it really was good for her. She loved it, she was well happy with it. (P7)

It's been great, [child] doesn't like other people, he doesn't like crowds and he doesn't want to be interacting with other people so the government saying to other people you can't interact with people I think was great, we enjoyed it. (P23)

A few parents suggested that the lockdown had actually improved their child's mental health.

He is much happier, much much happier during the pandemic. I wasn't worried about that side of things, he's happier in himself, he's more content, more confident, more self-belief in who he is as a person. He's got more resilient coming out of it. (P1)

However, some parents felt that their children's life skills and social skills had regressed due to the lack of contact with other children and were concerned about the impact this was having.

He's become a lot more withdrawn, I've seen quite a lot of regression in himself, his social skills. (P2)

### Subtheme 2.3: Strained relationships: 'Being constantly 24/7 together definitely did build up pressure'

For some families, staying physically safe had meant staying at home. Spending all their time together had

put more pressure on relationships within and between parents and children.

> A lot of shouting, particularly with a hormonal 15-year-old and a dad who's quite stressed, even though he's working from home he's probably doing longer hours than he was in the office, so it's been sort of little things that've built up and then there's been explosions. (P10)

Strategies that parents would normally use to de-escalate conflict, such as giving children time by themselves, were no longer possible when families could not leave the house for weeks at a time.

> The relationship between him and his brother has been very difficult during lockdown because we've not been able to give them their space from each other, so it's not been easy. (P19)

As the lockdown continued for several months (and for longer than many initially expected), many parents felt like they were unable to have a break and it took its toll on the parents' well-being.

> Now we're just on survival. (P19)

### Subtheme 2.4: Spending time together and slowing down: 'It brought us a lot closer together'

The main positive experience of lockdown, which was reported by the majority of parents, was the opportunity to spend more time together as a family as everyday life became less busy. Parents described having more time to be able to cook, eat and play together. Families felt closer and enjoyed their stronger relationships—something they wanted to continue after the lockdown finished.

> In terms of our relationship, it's had a positive effect because we're eating more family meals together and we're just generally spending more time together. (P13)

Being able to take things more slowly had a positive impact on well-being for both parents and children, as they felt freed from the daily rush. A break from rushing between different medical or school appointments was appreciated, with some parents noting that it was worth the switch to online services as they no longer had to take the time and energy to travel often significant distances for their child's appointments.

> Once we got in the swing of that, it was actually miles better because you didn't have to rush out for the bus, you know you're not working, you're not racing around trying to sort everything out. You can do everything at your own pace. (P9)

### Theme 3: Support matters

One factor which strongly contributed to the family's overall experience during the lockdown was the support provided by external sources. This includes educational support from schools, either for homeschooling or providing in-person schooling where this took place, as well as support for their child's medical and psychological needs. The amount and type of support provided by these services varied significantly, with some parents reporting regular contact and others no contact at all.

### Subtheme 3.1: Transition to telehealth 'I don't think you can replace face to face with a telephone'

Most children had complex behavioural and medical needs, which require regular clinical monitoring. Overall, 91% of families described the cancellation or postponement of routine medical and social care appointments.

> It's been really bad. All the appointments for the children have been cancelled. I've literally got a list on my fridge of about 15 appointments that I have to chase up, and get re-booked in once they can do so. (P14)

Most healthcare services started using telemedicine during the pandemic. Parents described varied experiences of telehealth. Some enjoyed the experience and were grateful that their child's appointments could still go ahead.

> They were quite good, they said they're not seeing anybody but they did again do it virtually. So they did a phone call and then I emailed pictures and stuff, which is quite funny trying to do that! But at least kind of that was, they're still trying to do it. (P12)

Many parents described needing specialist face-to-face care due to the complex nature of their child's needs and abilities. In some cases, it was challenging to communicate the complexity of their child's needs remotely.

> How can you explain to the physiotherapist how she walks over the phone? It's no good doing Zoom because obviously you still can't show, it's difficult so basically she's had to go off what I say and I've had to explain to her you know into detail and she done a report from there so that's difficult. (P16)

For others, the child was not able to communicate over the phone with professionals and so was not able to access the support that was offered.

> Yeah well we had a CAMHS [Child and Adolescent Mental Health Service] meeting but it was over Zoom and I just phoned them and I cancelled it, I said there's not a hope in hell you're going to get him to talk to you over a phone, he won't speak to anybody on the phone he doesn't, he can't get that there's somebody on the other side of something. (P23)

### Subtheme 3.2: Considering equality and equity in remote schooling
*Equality through access to technology: 'I think the first thing to do is make sure everyone is able to access what they're providing'*

Approximately half of children's schools remained open throughout the lockdown. Most children had at least a

few weeks of homeschooling and while some returned to in-person schooling partially or fully, other children remained at home until the lockdown ended in July. During the homeschooling period, schools made various attempts to adapt the educational experience to support parents and children, particularly supporting learning at home through online educational resources or teaching. Some parents reported that their school system provided them with resources to engage with online learning, which were appreciated.

> Anybody who was vulnerable or SEN got offered a tablet or a laptop. And most people took up on that. (Child) got his own laptop from school and that made a world of difference. (P8)

However, other families were not offered these same resources or the resources they were given were not appropriate or accessible. This often made it harder to engage with online learning.

> But everyone is in the same boat, and some children have to work on their phones because not everybody's got laptops, have they? And even doing on a tablet is not easy. So doing like math is okay on a tablet because you're just typing numbers but if you're like writing a longer piece of writing or a history essay on a phone or a tablet, that's a nightmare. (P9)

### *Equity of access lagging behind as adaptations aren't always appropriate*

However, many parents said that their child's level of intellectual disability, or other needs, meant that these resources were simply not accessible or required constant supervision from a parent.

> They developed a really fantastic online programme, but for the kids who have special needs that's very difficult to access. (P11)

Many parents felt that the level of work sent home was either too demanding or not challenging enough for their child. Children with IDD who were in mainstream education were often sent the same work as their typically developing peers and parents were offered little help in adapting this for their child's abilities, often having to rely on material that was inappropriate for their child's age.

> I think the biggest thing it would help it's special needs resources that aren't too babyish or patronising you know I'm getting work for him that meets his ability level academically but it's boring you know it's duckies on a pond and things like that. (P17)

### Subtheme 3: Checking in: 'just being able to have that support bubble, rather than being locked in your own four walls'

In general, parents felt the most supported when any support was targeted to the child's individual needs and when services were 'checking in' with parents, rather than waiting for problems to arise. Parents appreciated social and emotional support as much as, if not more than, practical support. This could come from individuals within services, or from friends and family, and was a way to let off steam when family life was difficult.

> I mean the other main thing to cope is – regularly calling up family or friends and having that contact. Because I'm with all the children I needed a break. So having that time to have a ten minute chat, to call, catch up. Tell someone how you're doing: 'I'm having a stressful day mum, how are you?' and those kind of things. This just made things easier. I think just talking to someone on the phone, an adult, as opposed to children. (P21)

Parents appreciated getting support from special educational needs services in school and felt better able to continue their child's education at home because they were able to ask questions and update the child's teachers on the amount of progress being made.

> We've had the SENCO [Special Educational Needs Co-ordinator] ringing us every three days checking in. She's also under the learning support service, that the school puts by for them and they support [child] for maths, but they've been ringing me every week. The head teacher has even phoned a couple of times. I know from speaking to people that that's not the norm, but we've felt very very much supported and I've found that I've been quite honest with them as well to be honest. (P22)

When parents did feel supported by medical and social services, this was usually the result of individuals reaching out, rather than a system-wide approach to support.

> I've also got a caseworker. And she's been fantastic. She's been there if you needed her, she's dropped masks off to me. We've had hand sanitisers dropped off. So she's been an absolute gem. If I ever needed anything, again shopping, she would be more than happy to have done it. So I've had her support well so I've been quite lucky on that. (P6)

### DISCUSSION

In addition to the challenges experienced by neurotypical families during the pandemic, parents of children with IDD have also had to manage the distinct challenges related to their children's complex needs. Overall, there was extensive variation in parents' reported experiences; some of these variations are likely to reflect the wide range of needs of children in the IMAGINE-ID cohort. Although all children in the cohort have IDD, some children attended mainstream schools and had relatively low levels of support needs, whereas others had multiple physical and behavioural needs.

## Parental distress

Overall, the parents displayed great resilience, but this often came at the detriment of their own mental health and well-being. In line with previous research, we found that caregivers had experienced increased levels of distress.[9 29–31 33] Where children needed high levels of personal care, parents often felt overwhelmed and unsupported, unable to access the respite care or specialised support they normally received. Many described high levels of distress and being close to burn-out, as confirmed by high levels of anxiety in the questionnaire. Evidence from other research also suggests that during the pandemic, the parents of children with IDD reported higher levels of burn-out than the parents of children without disabilities[47] and that the burden of care demands placed carers under increased strain.[48]

The findings also confirm previous reports that service provision for children with IDD had reduced or stopped during the pandemic.[4 5] Similar findings were observed in our study, with most parents reporting at least some cancelled medical or social appointments. Social support for the parents in our study was often limited, particularly for lone parents. Many parents felt that online and telephone communication with friends and family did not provide enough support and several felt isolated as their experience of parents of children with IDD was very different to that of other parents. Parents who reported feeling supported appeared to experience less distress. In a larger quantitative study, Willner *et al*[31] reported that the parents of people with an IDD received less support than parents of those without IDD, with those caring for people with severe challenging behaviour receiving the least support. This suggests that families who might have benefited most from support were the least likely to receive it.

## Short but regular check-ins

Regular check-ins with services and schools were described as very helpful, a novel insight that has not, to our knowledge, been reported in previous research. Parents who reported more positive experiences with online schooling tended to report frequent, individualised contact such as phone calls and emails with their children's teachers and support staff, which helped the parent adapt schoolwork for their child. A 5 min telephone call with each family once a week may be more effective than 2 hours working with the entire class, for children with special educational needs. Even when appointments were cancelled or moved online, frequent communication with services meant parents felt supported during this time of uncertainty. Frequent, brief check-ins are a cheap and easy way to maintain the support available to families during times of crisis and may reduce the risk of ongoing concerns developing into significant problems.

## Unexpected positives

Most families were able to draw some positives from the pandemic, as they were able to enjoy spending more time together, as has been previously reported.[4 9 33] Parents also appreciated stepping back from the rush of meetings and appointments, again suggesting that regular check-ins (which might be remote) might suit some families more than having to attend appointments in person.

## Impact on children

The impact of the pandemic on children's behaviour varied substantially. Some parents reported an increase in behavioural outbursts, while others explained that spending more time at home had led to a decrease in stress due to less demands on social interaction. A study by Neece *et al*[4] found that 'the parents of children with mental and physical disabilities were more likely to report changes in their child's behaviour, such as distractibility, inability to concentrate, irritability and general discomfort'. Similar findings were observed in the present study, with several parents reporting that behavioural challenges their children had previously outgrown re-emerged because of changes in routines and lack of support.

Many children were reported to benefit from routines and consistency in their daily lives; these were often disrupted at the start of the lockdown as schools and services closed down and many families started shielding. Where parents were able to introduce new routines, this seemed to benefit children, particularly when these routines allowed children more time at home doing activities they enjoyed. Establishing new routines may be a predictor of positive adjustment in this group. The need for routine has been identified in other research into the impact of the pandemic; lack of routine and or structured support has been associated with frustration for children, as reported by parents.[4 33]

## Hardware is a starting point, not a solution

Parents highlighted an important difference between the equality and equity of access to digital services. They were often given equality of access to technology, as school provided most families with laptops during the home-schooling period. However, the equity of access often lagged behind, as the resources made available to them in a digital format were not sufficiently tailored to allow their children to engage with them. This compliments the observation by Neece *et al*[4] that some parents felt ill-equipped to support their child's online learning. Without adaptations, digital homeschooling has the potential to increase inequalities. An individualised approach is essential to understand what is needed and the best ways to provide support.

## Telemedicine as complementary care delivery

Telemedicine is unlikely to be suitable for all IDD families. In the early stages of the pandemic, many medical and social services closed completely. However, by the time the interviews took place, some were starting to introduce telehealth services to keep in contact with the families in this study. Families enjoyed the flexibility that telemedicine gave them, reducing the need to travel to

appointments with children. However, there was a real sense that telehealth was not an appropriate substitute for in-person assessments, particularly for children with IDD. Parents felt that they were not able to communicate their children's often complex needs over the phone or email and the children themselves often struggled to engage with medical professionals who they could not see in front of them. These issues have been previously raised with regard to telehealth services provided during the pandemic, with the need for adaptation highlighted.[49 50] While telemedicine offers many opportunities to expand services, it is important to emphasise that this new mode of healthcare provision will not suit everyone and families of children with IDD should be offered a range of options to ensure their child receives appropriate support. Reflecting a broader shift to digital services across healthcare provision, blended care will likely be the most appropriate approach going forward.

### Strengths, limitations and future directions

The study reflected the experiences of the mothers who were interviewed. Clearly, this small study cannot represent the full extent of the impact of the first UK lockdown on the IDD community. Our use of consecutive sampling was necessitated by the approaching end of shielding in the UK in August 2020; we felt it was important to interview all participants under the same level of restrictions. However, this also means that we were not able to selectively recruit participants and so it is unclear to what extent variations in experience are related to variations in child's level of need or additional factors. Our interviewees were all female and were mostly white and in co-parenting relationships; experiences are likely to differ for non-white, non-female, single parents.

Concurrent large-scale questionnaire-based studies, such as those conducted by Willner *et al*,[31] are needed to address this. Small-scale qualitative studies can identify broader experiences than might be reported in large-scale quantitative studies and can produce novel hypotheses for further examination. However, a key limitation of our methodology is its lack of generalisability outside of the first UK lockdown, to other countries or timepoints with different levels of infection and restrictions. We also did not interview any children or young people with IDD and therefore our findings are reliant on parents' perceptions of their child's experience, rather than firsthand accounts. Future research should seek to explore the experiences of children with IDD through their own communication, as there are undoubtedly features which have been missed in the present study.

The interviews contained a few notable absences. We had anticipated reports of excessive hand washing and cleanliness, however this was rarely mentioned in our interviews. We had also expected some families to express concern about being judged for needing allowances/exemptions, such as their child not wearing a mask when out in public, but this was rarely discussed. No one reported breaking the rules. Although this could

be attributed to expectancy biases, we interpret this to be representative of the high national compliance during the first UK lockdown, particularly in individuals with preexisting medical conditions.[51] Research by Rogers *et al*[33] with the mothers of IDD children highlighted themes of stigma and powerlessness, which were not as prominent in our interviews.

The views presented here reflect families' experiences in the first few months of the pandemic and the first UK lockdown, a time of particularly heightened anxieties and stringent restrictions. At the time of writing the manuscript, the government's response to the pandemic continues to evolve monthly. It is likely that parents' experiences and attitudes have changed over time. Therefore, we plan to conduct follow-up interviews in the spring of 2021, which may offer different insights as families have had more time to adapt to the 'new normal'. This will also allow us to explore concerns about how children with IDD have navigated the changes in rules[25–35] and explore the impact of a prolonged national crisis on the carers of children with IDD.

### CONCLUSION

Our findings confirm observations previously found in UK parents of children with IDD[33] and provide new insights on the use of technology during the pandemic for schooling and healthcare, as well as the need for regular check-ins. Parents of children with IDD experienced many challenges during the UK COVID-19 outbreak during the spring/summer of 2020. In addition to the demands of homeschooling, concern over infection and lack of social connection that families across the country dealt with, families in this study had to meet the additional and often complex educational and medical needs of their children. Many parents felt let down and forgotten by medical and school services and felt isolated from the rest of the country. However, parents also used their experience to develop coping strategies and found some unexpected benefits for their children, such as spending more time together at home. We can learn from these parents' experiences to provide guidelines for the future, to ensure these families are not forgotten.

**Acknowledgements** The authors would like to thank all the families who took part in this study and the IMAGINE-ID consortium for making this research possible.

**Collaborators** IMAGINE-ID Consortium: Kate Baker, Eleanor Dewhurst, Amy Lafont, F Lucy Raymond, Terry Shirley, Hayley Tilley, Husne Timur, Catherine Titterton, Neil Walker, Sarah Wallwork, Francesca Wicks, Zheng Ye, Marie Erwood, Sophie Andrews, Philippa Birch, Samantha Bowen, Karen Bradley, Aimee Challenger, Samuel Chawner, Andrew Cuthbert, Jeremy Hall, Peter Holmans, Sarah Law, Nicola Lewis, Sinead Morrison, Hayley Moss, Michael Owen, Sinead Ray, Matthew Sopp, Molly Tong, Marianne van den Bree, Nadia Coscini, Sarah Davies, Spiros Denaxas, Hayley Denyer, Nasrtullah Fatih, Manoj Juj, Ellie Kerry, Anna Lucock, Frida Printzlau, Ramya Srinivasan, Susan Walker, Alice Watkins, Beverly Searle, Anna Pelling, John Dean, Lisa Robertson, Denise Williams, Alan Donaldson, Annie Procter, Jonathan Berg, Anne Lampe, Julia Rankin, Shelagh Joss, Lyn Chitty, Frances Flinter, Muriel Holder, Alison Kraus, Julian Barwell, Pradeep Vasudevan, Astrid Weber, William Newman, Miranda Splitt, Virginia Clowes, Fleur van Dijk, Rachel Harrison, Usha Kini, Oliver Quarrell, Diana Baralle, Sahar Mansour and Yanick Crow.

**Contributors** JW, LH and DS designed and directed the study. JW and LH conducted the interviews and analysis (theme development and review). Themes were reviewed following several detailed discussions with the other members of the research team DS and WM. Transcripts were coded by independent reviewers LW and TNA. The IMAGINE-ID consortium contributed to recruitment and manuscript revisions.

**Funding** The IMAGINE-ID study was funded by the Medical Research Council and Medical Research Foundation (MR/L011166/1 and MR/N022572/1). Research conducted at the University College London Great Ormond Street Institute of Child Health was supported by the National Institute of Health Research Biomedical Research Centre. The funders played no role in the design of the study and collection, analysis, interpretation of data or writing of the manuscript.

**Competing interests** None declared.

**Patient consent for publication** Obtained.

**Ethics approval** Ethical approval for the study was obtained from UCL Research Ethics Committee (9553/003).

**Provenance and peer review** Not commissioned; externally peer reviewed.

**Data availability statement** Data are available upon reasonable request. The datasets generated and/or analysed during the current study are not publicly available in order to protect the anonymity of those taking part, but are available from the corresponding author on reasonable request.

**ORCID iDs**
Jeanne Wolstencroft http://orcid.org/0000-0001-6160-9731
Tooba Nadeem Akhtar http://orcid.org/0000-0003-0246-7299

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
