## [Reviewer comments · BMJ Open]

ARTICLE DETAILS

TITLE (PROVISIONAL)	"We've been in lockdown since he was born": A mixed-methods exploration of the experiences of families caring for children with intellectual disability during the Covid-19 pandemic in the UK
AUTHORS	Wolstencroft, Jeanne; Hull, Laura; Warner, Lauren; Akhtar, Tooba Nadeem; Mandy, W; IMAGINE, ID; Skuse, David

VERSION 1 – REVIEW

REVIEWER	Marsh, Lynne Queen's University, School of Nursing and Midwifery
REVIEW RETURNED	08-Mar-2021

GENERAL COMMENTS	This is a well written and topical study and adds to the wider research. It was informative and presented clear recommendations and future considerations. The title captures the reader from the outset, and it was an excellent portrayal of one mother's experience of her life with her child with an intellectual disability and was insightful. I appreciate that this was not all mothers experiences, but still added to a better understanding of what life may be for some families outside of the Covid pandemic. The voices of the mums were clearly identified resulting in a balanced discussion. It was delightful to hear that their experiences were not all negative and there was some positive aspects to this pandemic. There is learning from this experience and well done for raising an awareness of the experiences of mum's and their families.
--

REVIEWER	Provenzi, Livio Scientific Institute IRCCS E. Medea, 0-3 Center for the at-Risk Infant
REVIEW RETURNED	15-Mar-2021

GENERAL COMMENTS	Dear authors, thanks for sending your manuscript. I read your study carefully and I do believe that exploring and reporting on the psychological experience of parents of children with IDD is very relevant during the healthcare emergency that we are experiencing. Nonetheless, this study has too many limitations. Some refer to the manuscript and require the authors to better organize the paper. Other limitations reflect methodological problems that largely undermine the reliability and rigor of this study. Finally, there is a large omission of citations of previous literature on this topic; despite the authors state that there is no previous publication, this seems to be wrong. Please, find below my specific concerns.
---

	1. In the introduction, the sentences appear to be juxtaposed and the lack of logical connectives makes the reading less fluid than optimal. Also, the authors state that “To our knowledge, there has not yet been any research specifically examining the experiences of caregivers of children with IDD, who have been caring for their children at home during the pandemic”. Nonetheless, this is not correct. Please, cite appropriate literature (here we list some examples – but others may be available):  • Cacioppo, M., Bouvier, S., Bailly, R., Houx, L., Lempereur, M., Mensah-Gourmel, J., ... & ECHO Group. (2020). Emerging health challenges for children with physical disabilities and their parents during the COVID-19 pandemic: The ECHO French survey. Annals of physical and rehabilitation medicine. • Cahapay, M. B. (2020). How Filipino parents home educate their children with autism during COVID-19 period. International Journal of Developmental Disabilities, 1-4. • Grumi, S., Provenzi, L., Gardani, A., Aramini, V., Dargenio, E., Naboni, C., ... & Engaging with Families through On-line Rehabilitation for Children during the Emergency (EnFORCE) Group. (2021). Rehabilitation services lockdown during the COVID-19 emergency: the mental health response of caregivers of children with neurodevelopmental disabilities. Disability and Rehabilitation, 43(1), 27-32. • Neece, C., McIntyre, L. L., & Fenning, R. (2020). Examining the impact of COVID-19 in ethnically diverse families with young children with intellectual and developmental disabilities. Journal of Intellectual Disability Research, 64(10), 739-749. 2. “Recommendations have been developed from these” is vague: please, be more precise when describing your rationale. 3. “At this time, the lockdown restrictions were eased in parts of the UK, but the government advised vulnerable individuals to continue to shield until the 1st of August 2020.  This meant that families were able to reflect on their experiences of the first national lockdown”. It is not clear to me why the authors believe that there is a consequential connection between these two sentences. 4. Please, consider organizing the findings in a better way. For example, separate strengths and challenges more clearly in two paragraphs. In the present version of the manuscript, the findings related to “support” are widely distributed and not presented in specific sub-paragraphs – this prevents the reader to develop a coherent understanding of the findings. 5. Also, for what pertains to the aims, the second one is poorly defined (the expression “to understand more” is too vague) and the third one is an implication of this study but does not sound like a proper study aim. I would suggest removing aim 3 (all studies have implications). Otherwise, if this study was meant to build from these interviews to provide recommendations, there is evidence that appropriate methodology was used (e.g., Delphi? Nominal groups?). 6. The sample is too heterogeneous. In qualitative research, the subjects should be selected through purposive sampling – not consecutive one. In this case, the presence of large individual differences – e.g., family structure, general developmental characteristics of the children – makes it difficult to understand if the findings were typical of a specific population. In other words, based on the heterogeneity of the sample, one would wonder if the themes may change in families of children with different characteristics. Nonetheless, a QL study with 23 families does not seem to be well suited to respond to this goal. Moreover, the
--	--

	authors report in detail information that is not relevant for this study (e.g., ethnicity): do the authors believe that ethnicity may impact the findings? If yes, how? This is not described in the introduction. 7. Finally, the clinical diagnoses of the children should be more clearly reported as many readers may not be familiar with the chromosomal variants reported in Table S2. I would suggest including a Table that provides a more complete description of each family and of each child: age, gender, socio-economic status, parents' educational level (years of study), main diagnosis, psychomotor delay indexes. In the absence of this information, it is very difficult to understand who these families were and how this study may inform practice for families with children diagnosed with IDD. 8. Also, please check the match between Table/Figure captions (S2 is indexed as S1, page 5). 9. The description of the QL methodology adopted is quite completely missing. Which approach was used for the interviews? Was it a phenomenological approach? Did the authors use IPA interviews? The description of the qualitative methodology is largely unsatisfying. 10. "Overall, parents described experiencing similar difficulties to those they observed in other families during the lockdown, but with the additional challenge of their child's specific needs". this sentence does not seem to be supported by the findings as there was no comparison between the experience of families with IDD-diagnosed children and families of healthy/typically developing children. 11. Also, the quotations should be identified with an ID: I could not understand how many quotations were obtained from the same interview. Consistently, the results section is too messy. 12. Confusion also arises for what pertains to thematic labels. Under the label "support matters" the authors include dimensions related to parents' feelings of support as well as parents' efforts to support children in school. Nonetheless, by doing so, they merge two very different topics that should be discussed separately. Moreover, the section related to healthcare support is reported in another thematic cluster (i.e., "managing pre-existing challenges"). 13. The labels for themes are sometimes too long and do not completely adhere to the examples reported from the interviews. Also, for some of the thematic clusters (e.g., "nice but hard") there are no examples reported. 14. In Figure 1, the authors report a "thematic map of findings". This figure does not seem a map. Is more like a list of themes. Also, as this study seems to be more like a phenomenological QL approach (but not properly described), creating a map or a theoretical big picture seems to be inappropriate. QL approaches that are meant to develop a theoretical model (that is what I imagine by reading the term "thematic map") are more suited for grounded theory approaches. Nonetheless, the present study does not seem to be a ground theory. Also, in the "thematic map" there are three different colors: what do they refer to? 15. The discussion is disproportionate. The authors provide a very limited and brief discussion of the findings. At the same time, they largely focus on the implications, that seems to be only partially supported by the results. Also (see comment #1), the discussion largely disregards the rapidly accumulating literature on the psychological response of parents of children with IDD during the COVID-19 pandemic (for the UK context: Rose et al. (2020) – Int J Dev Disabilities).
--	---

REVIEWER	Willner, Paul University of Wales Swansea
REVIEW RETURNED	29-Mar-2021

GENERAL COMMENTS	At the time of writing this was the first study to explore experiences of parents of children with IDD during the first UK lockdown. However, another study has very recently been published (Rogers et al, 2021, The experiences of mothers and young children with intellectual disabilities during the first COVID-19 lockdown period: JARID, http://doi.org/10.1111/jar.12884. (This was a subset of a sample of parents whose quantitative data were reported in Willner et al., 2020, JARID 33, 1523-33.) The authors will need to change the claim of originality in the summary and incorporate some discussion of their findings in relation to this earlier work, which was conducted closer to the end of the lockdown. To my reading, the two samples have quite similar demographics and the two sets of data are similar (as would be expected) but not identical. More detail of the interviews is needed, including the questions asked and whether parents were responding about their experiences at the time, or were asked explicitly to reflect back on the earlier period of strict lockdown. (The full interview schedule is provided as supplementary material but more information should be included in the main text.) Also, the length of the interview should be reported. Fig. 1 presents a clear overview of the themes extracted from the interviews, but the text is much more difficult to navigate. Differential use of headers and sub-headers would help to align the text to the overview. I also wonder whether the Results section might benefit from shortening to focus on those areas that feed into the issues highlighted in the Discussion. One point that struck me when comparing this Discussion to the earlier paper by Rogers et al was that the areas that receive greater prominence here (predominantly thoughts about IT and telemedicine) perhaps relate to an area that had developed over the period between the two studies. Another notable difference that may merit comment was the presence there but absence here (albeit expected) of reports of stigma. The Conclusion and Recommendations are clear and important. Minor comments: p.5, l.52: Delete “for anxiety symptoms” p.15, l.58: “high” not “increased” p.16, l.3: Please refer here to Willner et al., 2020, who made the same point on the basis of a large quantitative sample. Overall, this is a clear and well-presented (if perhaps over-long) report that adds weight to and strengthens the recommendations of earlier work.
--

VERSION 1 – AUTHOR RESPONSE

Reviewer: 1

Dr. Lynne Marsh, Queen's University

Comments to the Author:

This is a well written and topical study and adds to the wider research. It was informative and presented clear recommendations and future considerations. The title captures the reader from the outset, and it was an excellent portrayal of one mother's experience of her life with her child with an intellectual disability and was insightful. I appreciate that this was not all mothers experiences, but still added to a better understanding of what life may be for some families outside of the Covid pandemic. The voices of the mums were clearly identified resulting in a balanced discussion. It was delightful to hear that their experiences were not all negative and there were some positive aspects to this pandemic. There is learning from this experience and well done for raising an awareness of the experiences of mums and their families.

We thank you for your kind comments.

Reviewer: 2

Dr. Livio Provenzi, Scientific Institute IRCCS E. Medea

Comments to the Author:

Dear authors, thanks for sending your manuscript.

I read your study carefully and I do believe that exploring and reporting on the psychological experience of parents of children with IDD is very relevant during the healthcare emergency that we are experiencing. Nonetheless, this study has too many limitations. Some refer to the manuscript and require the authors to better organize the paper. Other limitations reflect methodological problems that largely undermine the reliability and rigor of this study. Finally, there is a large omission of citations of previous literature on this topic; despite the authors state that there is no previous publication, this seems to be wrong.

We thank you for your comments and have addressed your comments in turn below.

Please, find below my specific concerns.

1. In the introduction, the sentences appear to be juxtaposed and the lack of logical connectives makes the reading less fluid than optimal. Also, the authors state that "To our knowledge, there has not yet been any research specifically examining the experiences of caregivers of children with IDD, who have been caring for their children at home during the pandemic". Nonetheless, this is not correct. Please, cite appropriate literature (here we list some examples – but others may be available):

- Cacioppo, M., Bouvier, S., Bailly, R., Houx, L., Lempereur, M., Mensah-Gourmel, J., ... & ECHO Group. (2020). Emerging health challenges for children with physical disabilities and their parents during the COVID-19 pandemic: The ECHO French survey. *Annals of physical and rehabilitation medicine*.

- Cahapay, M. B. (2020). How Filipino parents home educate their children with autism during COVID-19 period. *International Journal of Developmental Disabilities*, 1-4.
- Grumi, S., Provenzi, L., Gardani, A., Aramini, V., Dargenio, E., Naboni, C., ... & Engaging with Families through On-line Rehabilitation for Children during the Emergency (EnFORCE) Group. (2021). Rehabilitation services lockdown during the COVID-19 emergency: the mental health response of caregivers of children with neurodevelopmental disabilities. *Disability and Rehabilitation*, 43(1), 27-32.
- Neece, C., McIntyre, L. L., & Fenning, R. (2020). Examining the impact of COVID-19 in ethnically diverse families with young children with intellectual and developmental disabilities. *Journal of Intellectual Disability Research*, 64(10), 739-749.

We have updated the introduction to include more recent references and to improve the flow. At the time of writing none of the references in European or UK settings were available.

2. "Recommendations have been developed from these" is vague: please, be more precise when describing your rationale.

This has been amended and the rationale has been made more explicit.

3. "At this time, the lockdown restrictions were eased in parts of the UK, but the government advised vulnerable individuals to continue to shield until the 1st of August 2020.  This meant that families were able to reflect on their experiences of the first national lockdown". It is not clear to me why the authors believe that there is a consequential connection between these two sentences.

We have made it clearer that we asked participants to look back at their experiences of the pandemic, and that we conducted interviews when all participants were being given the same advice and following the same restrictions.

4. Please, consider organizing the findings in a better way. For example, separate strengths and challenges more clearly in two paragraphs. In the present version of the manuscript, the findings related to "support" are widely distributed and not presented in specific sub-paragraphs – this prevents the reader to develop a coherent understanding of the findings.

We appreciate the reviewer's request to organise the findings more clearly; we have restructured themes and subthemes following all reviewers' very helpful comments. However, we were hesitant to separate 'strengths' and 'challenges' into two separate themes, as for most of the families we spoke to, their experience combined both strengths and challenges at the same time, as a result of the broader themes we identified (such as enjoying spending time together as a family, but missing out on wider social support because of this). We have therefore kept these and the findings related to support spread across the different themes, and have instead revised the discussion to bring together some broader conclusions related to strengths, challenges, and support.

5. Also, for what pertains to the aims, the second one is poorly defined (the expression "to understand more" is too vague) and the third one is an implication of this study but does not sound like a proper study aim. I would suggest removing aim 3 (all studies have implications). Otherwise, if this study was meant to build from these interviews to provide recommendations, there is evidence that appropriate methodology was used (e.g., Delphi? Nominal groups?).

The aims have been updated as advised.

6. The sample is too heterogeneous. In qualitative research, the subjects should be selected through purposive sampling – not consecutive one. In this case, the presence of large individual differences – e.g., family structure, general developmental characteristics of the children – makes it

difficult to understand if the findings were typical of a specific population. In other words, based on the heterogeneity of the sample, one would wonder if the themes may change in families of children with different characteristics. Nonetheless, a QL study with 23 families does not seem to be well suited to respond to this goal. Moreover, the authors report in detail information that is not relevant for this study (e.g., ethnicity): do the authors believe that ethnicity may impact the findings? If yes, how? This is not described in the introduction.

As you rightly highlight, children in the IMAGINE cohort are wonderfully diverse, however their parents share the experience of caring for a child with special needs. In our experience of working with these families for the last 7 years, this is meaningful common ground. We chose to use qualitative methods in this study as we were interested in capturing the experiences and perspective of caregivers during the pandemic, this aim could not have been addressed without qualitative research methods. Furthermore, our previous research has also shown the medical and behavioural complexity of our cohort, a complexity which could not be contextualised without an in-depth qualitative approach.

The themes would most certainly change if we had interviewed the parents of children with different characteristics, as is true of all thematic analyses. The themes would also have been different if the participants had been 'matched' or if the sample had been more homogeneous. The aim of our study was not to reach an 'objective truth', but rather, to gain insights into the experiences of the families who took part in our study.

We do not know whether ethnicity would have an impact on our findings, future research would be needed to explore this. You signposted us to a reference above (Neece et al. 2020), which highlights that research on the impact of ethnically diverse groups during the pandemic is scarce. We have included a brief description of the ethnicity of our participants as it is important to acknowledge that our sample is biased towards white participants. We appreciate that our use of consecutive sampling likely impacted the characteristics of participants; due to time restrictions (interviewing all participants before the end of shielding recommendations in the UK in August 2020) we were not able to use purposive sampling. We have now acknowledged this in the methods and discussion of the paper.

7. Finally, the clinical diagnoses of the children should be more clearly reported as many readers may not be familiar with the chromosomal variants reported in Table S2. I would suggest including a Table that provides a more complete description of each family and of each child: age, gender, socio-economic status, parents' educational level (years of study), main diagnosis, psychomotor delay indexes. In the absence of this information, it is very difficult to understand who these families were and how this study may inform practice for families with children diagnosed with IDD.

Thank you for this suggestion. We have created a table to summarise the participant characteristics.

8. Also, please check the match between Table/Figure captions (S2 is indexed as S1, page 5).

This has been updated.

9. The description of the QL methodology adopted is quite completely missing. Which approach was used for the interviews? Was it a phenomenological approach? Did the authors use IPA interviews? The description of the qualitative methodology is largely unsatisfying.

As described in the methods section we took a critical realist framework approach to reflective thematic analysis. We have expanded on our theoretical stance and also added a paragraph on our reflective practice process.

10. “Overall, parents described experiencing similar difficulties to those they observed in other families during the lockdown, but with the additional challenge of their child’s specific needs”. This sentence does not seem to be supported by the findings as there was no comparison between the experience of families with IDD-diagnosed children and families of healthy/typically developing children.

Although our study did not compare the experiences of our group to a control group or other families, families often compared themselves to the families of children with ‘neurotypical children’, therefore we feel it is valid to include this theme, with the emphasis that this was parents’ own perceptions.

11. Also, the quotations should be identified with an ID: I could not understand how many quotations were obtained from the same interview. Consistently, the results section is too messy.

We have added IDs to the quotes.

12. Confusion also arises for what pertains to thematic labels. Under the label “support matters” the authors include dimensions related to parents’ feelings of support as well as parents’ efforts to support children in school. Nonetheless, by doing so, they merge two very different topics that should be discussed separately.

We feel these two constructs are interlinked and need to be discussed together, as well supported parents were better able to support their children (as we note in the discussion). We have changed the heading to differentiate these two aspects of support (parent – child) in order to make it clearer.

Moreover, the section related to healthcare support is reported in another thematic cluster (i.e., “managing pre-existing challenges”).

We have now restructured the themes to include the sub-theme telehealth within the broader context of support, whereas the ‘pre-existing challenges’ theme refers to behavioural issues and issues around communicating about Covid with children with IDD. As the latter theme does not relate directly to support from other services, we have not integrated telehealth with this.

13. The labels for themes are sometimes too long and do not completely adhere to the examples reported from the interviews. Also, for some of the thematic clusters (e.g., “nice but hard”) there are no examples reported.

We have edited some of the theme labels to make them shorter. In this specific example, instance “nice but hard” is the key quote; we have amended the theme header to make this clearer, and have provided example quotations for all the sub-themes within this. We recognise that the quotations within theme labels were sometimes long, as we did not want to edit participants’ quotations to change the meaning or intention of their experiences. We have therefore kept some longer quotations to ensure participants’ voices remain distinct.

14. In Figure 1, the authors report a “thematic map of findings”. This figure does not seem a map. It is more like a list of themes. Also, as this study seems to be more like a phenomenological QL approach (but not properly described), creating a map or a theoretical big picture seems to be inappropriate. QL approaches that are meant to develop a theoretical model (that is what I imagine by reading the term “thematic map”) are more suited for grounded theory approaches. Nonetheless, the present study does not seem to be a grounded theory. Also, in the “thematic map” there are three different colors: what do they refer to?

Figure 1 is intended to give an overview of the themes rather than a theoretical model and has been renamed to “Summary of themes and sub-themes” for clarity. As Reviewer 3 reported that Figure 1 was helpful, we have kept this table of themes to give readers an overall summary of the findings.

15. The discussion is disproportionate. The authors provide a very limited and brief discussion of the findings. At the same time, they largely focus on the implications, that seems to be only partially supported by the results. Also (see comment #1), the discussion largely disregards the rapidly accumulating literature on the psychological response of parents of children with IDD during the COVID-19 pandemic (for the UK context: Rose et al. (2020) – Int J Dev Disabilities).

At time of writing this literature was unavailable, we have now updated our manuscript to reflect the literature which has been published since submission, and have extended the discussion to account for this and for the contribution our study adds to the recommendations we suggest.

Reviewer: 3

Dr. Paul Willner, University of Wales Swansea

We thank you for your constructive comments and congratulate you on your recent publications on the experiences of mothers caring for children with IDD. Comparing our findings has been most insightful.

Comments to the Author:

At the time of writing this was the first study to explore experiences of parents of children with IDD during the first UK lockdown. However, another study has very recently been published (Rogers et al, 2021, The experiences of mothers and young children with intellectual disabilities during the first COVID-19 lockdown period: JARID, <https://eur01.safelinks.protection.outlook.com/?url=http%3A%2F%2Fdoi.org%2F10.1111%2Fjar.12884&data=04%7C01%7C%7Cae8566ad5f2147efeab208d903ea2ef8%7C1faf88fea9984c5b93c9210a11d9a5c2%7C0%7C0%7C637545126629119352%7CUnknown%7CTWFpbGZsb3d8eyJWIjoiMC4wLjAwMDAiLCJQIjoiV2luMzliLCJBTil6lk1haWwiLCJXVCi6Mn0%3D%7C1000&data=TTN89i3ytFE1E8uMVC34y%2FrAtSZDqdyLkZfrJ2W9cWs%3D&reserved=0>. (This was a subset of a sample of parents whose quantitative data were reported in Willner et al., 2020, JARID 33, 1523-33.) The authors will need to change the claim of originality in the summary and incorporate some discussion of their findings in relation to this earlier work, which was conducted closer to the end of the lockdown. To my reading, the two samples have quite similar demographics and the two sets of data are similar (as would be expected) but not identical.

More detail of the interviews is needed, including the questions asked and whether parents were responding about their experiences at the time, or were asked explicitly to reflect back on the earlier period of strict lockdown. (The full interview schedule is provided as supplementary material but more information should be included in the main text.) Also, the length of the interview should be reported.

These suggestions have been implemented and the methods section has made these point clearer. In particular, we have clarified that parents were asked to reflect back on their experiences in earlier periods of strict lockdown, as well as their current experiences at the time the interview was conducted.

Fig. 1 presents a clear overview of the themes extracted from the interviews, but the text is much more difficult to navigate. Differential use of headers and sub-headers would help to align the text to the overview. I also wonder whether the Results section might benefit from shortening to focus on those areas that feed into the issues highlighted in the Discussion.

We have amended our use of headers to make it align more clearly to the overview. We have also shortened the results section to focus on the issues brought up in the discussion.

One point that struck me when comparing this Discussion to the earlier paper by Rogers et al was that the areas that receive greater prominence here (predominantly thoughts about IT and telemedicine) perhaps relate to an area that had developed over the period between the two studies. Another notable difference that may merit comment was the presence there but absence here (albeit expected) of reports of stigma.

Thank you for this observation. We have updated our discussion to highlight these notable differences.

The Conclusion and Recommendations are clear and important.

Minor comments:

p.5, l.52: Delete “for anxiety symptoms” - This has now been deleted.

p.15, l.58: “high” not “increased” – this has been amended.

p.16, l.3: Please refer here to Willner et al., 2020, who made the same point on the basis of a large quantitative sample. - This has been amended.

Overall, this is a clear and well-presented (if perhaps over-long) report that adds weight to and strengthens the recommendations of earlier work.